# Crystal Structure and Properties of Gd$_{1-x}$Sr$_x$Co$_{1-y}$Fe$_y$O$_{3-\delta}$ Oxides as Promising Materials for Catalytic and SOFC Application

**Tatiana V. Aksenova \*, Darya K. Mysik and Vladimir A. Cherepanov** 

Institute of Natural Science and Mathematics, Ural Federal University, 620000 Yekaterinburg, Russia
\* Correspondence: TV.Aksenova@urfu.ru

**Abstract:** A series of samples with the overall composition Gd$_{1-x}$Sr$_x$Co$_{1-y}$Fe$_y$O$_{3-\delta}$ ($x$ = 0.8; 0.9 and $0.1 \leq y \leq 0.9$), which are promising materials for catalytic and SOFC application, was prepared by a glycerol nitrate technique. X-ray diffraction analysis allowed to describe Gd$_{0.2}$Sr$_{0.8}$Co$_{1-y}$Fe$_y$O$_{3-\delta}$ with $0.1 \leq y \leq 0.5$ in a tetragonal $2a_p \times 2a_p \times 4a_p$ superstructure (SG $I4/mmm$), while oxides with $0.6 \leq y \leq 0.9$ exhibit cubic disordered perovskite structure (SG $Pm$-3$m$). All Gd$_{0.1}$Sr$_{0.9}$Fe$_{1-y}$Co$_y$O$_{3-\delta}$ oxides within the composition range $0.1 \leq y \leq 0.9$ possess the cubic perovskite structure (SG $Pm$-3$m$). The structural parameters were refined using the Rietveld full-profile method. The changes of oxygen content in Gd$_{1-x}$Sr$_x$Co$_{1-y}$Fe$_y$O$_{3-\delta}$ versus temperature were determined by thermogravimetric analysis. The introduction of iron into the cobalt sublattice leads to a gradual increase in the unit cell parameters and unit cell volume, accompanied with increasing oxygen content. The temperature dependency of conductivity for Gd$_{0.2}$Sr$_{0.8}$Co$_{0.3}$Fe$_{0.7}$O$_{3-\delta}$ exhibits a maximum (284 S/cm) at ≈600 K in air. The positive value of the Seebeck coefficient indicates predominant $p$-type conductivity in the Gd$_{0.2}$Sr$_{0.8}$Co$_{0.3}$Fe$_{0.7}$O$_{3-\delta}$ complex oxide.

**Keywords:** solid solution; X-ray diffraction; crystal structure; thermogravimetric analysis; oxygen nonstoichiometry; conductivity

## 1. Introduction

The study of Sr-substituted rare earth cobaltites/ferrites is of great interest, since these materials can be used as catalysts [1–4], cathodes in solid oxide fuel cells (SOFCs) [5–10], as magnetic materials [11,12], or gas sensors [13–15]. Mixed Fe/Co occupation of B-sites in perovskite structure promotes a good compromise between the catalytic activity and phase stability of complex oxides [16–19]. The homogeneity ranges, crystal structure, and physicochemical properties of Ln$_{1-x}$Sr$_x$CoO$_{3-\delta}$ (Ln = lanthanide ion) were the subject of numerous studies [20–35]. It was shown that both the structure and properties depend on the size of the Ln cations and the concentration of the dopant (Sr). On the other hand, it is widely acknowledged that the variable oxidation state of Co and Fe ions and the concentration of oxygen vacancies play an important role in the catalytic activity of materials that are considered as catalysts for various Red-Ox processes.

The introduction of strontium into the A-sublattice in LnCoO$_{3-\delta}$, containing a large-size lanthanide (Ln = La, Pr, Nd), leads to the formation of extended ranges of solid solutions of various structures. Cobaltites La$_{1-x}$Sr$_x$CoO$_{3-\delta}$ with $0.0 \leq x \leq 0.5$ crystallized in a rhombohedral structure (SG $R$-3$c$) and further Sr substitution $0.6 \leq x \leq 0.8$ lead to a cubic structure (SG $Pm$-3$m$) [20–22]. The introduction of Sr into Ln$_{1-x}$Sr$_x$CoO$_{3-\delta}$ (Ln = Pr, Nd) within $0.0 \leq x \leq 0.5$ preserves an orthorhombically distorted perovskite-type structure (SG $Pbnm$) [23–25]. An increase in the Sr-content leads to a change in the structure from orthorhombic to cubic [24].

A significant oxygen deficiency in Sr-enriched Ln$_{1-x}$Sr$_x$CoO$_{3-\delta}$ (Ln = La-Nd) solid solutions prepared in oxygen flow and cooled to room temperature leads to the ordering of

oxygen vacancies along the *c*-axis and the formation of $a_p \times a_p \times 2a_p$ superstructure (SG $P4/mmm$), where $a_p$ is the unit cell parameter of the primitive perovskite cell [24,26,27].

The increased difference between ionic radii of Sr and Ln for the rare earth elements smaller than Nd (Ln = Sm, Gd) prevents the formation of Ln-enriched $Ln_{1-x}Sr_xCoO_{3-\delta}$ solid solutions with the orthorhombic structure [28–30]. At the same time, Sr-enriched oxides form wide ranges of solid solutions with $0.5 \leq x \leq 1.0$ for Ln = Sm and $0.6 \leq x \leq 1.0$ for Ln = Gd, which possess a tetragonal $2a_p \times 2a_p \times 4a_p$ superstructure (SG $I4/mmm$) [28–30]. It is worth noting that a similar cationic ordering was described by Istomin et al. [31] in terms of the 314-type structure ($Sr_3LnCo_4O_{12-4\delta}$ or $Sr_{0.75}Ln_{0.25}CoO_{3-\delta}$) for Ln = Y, Sm-Tm, with possible substitution of Ln by Sr or vice versa. It was found that the intensity of XRD peaks corresponding to the superstructure in $Ln_{1-x}Sr_xCoO_{3-\delta}$ decreases with an increase of strontium content [27,29,30].

The structural model of a tetragonal $2a_p \times 2a_p \times 4a_p$ unit cell for the $Ln_{1-x}Sr_xCoO_{3-\delta}$ complex oxide is shown in Figure 1. It includes three different A-sites, which are successively filled with Ln ions as their total content increases. According to James et al. [27], at the first stage, Ln ions exclusively fill only A1-sites, while A2- and A3-sites remain completely occupied by $Sr^{2+}$ ions. A further increase in the Ln content ($x > 0.25$) is carried out by replacing strontium with lanthanide in A3-sites, while A2 remain fully occupied by $Sr^{2+}$ ions. Although the general approach used by Istomin et al. [31] for the distribution of Sr and Ln at three different A-sites was the same, they used opposite designations for A2 and A3 sites and the opposite location of Sr and Ln in Ln-rich oxides.

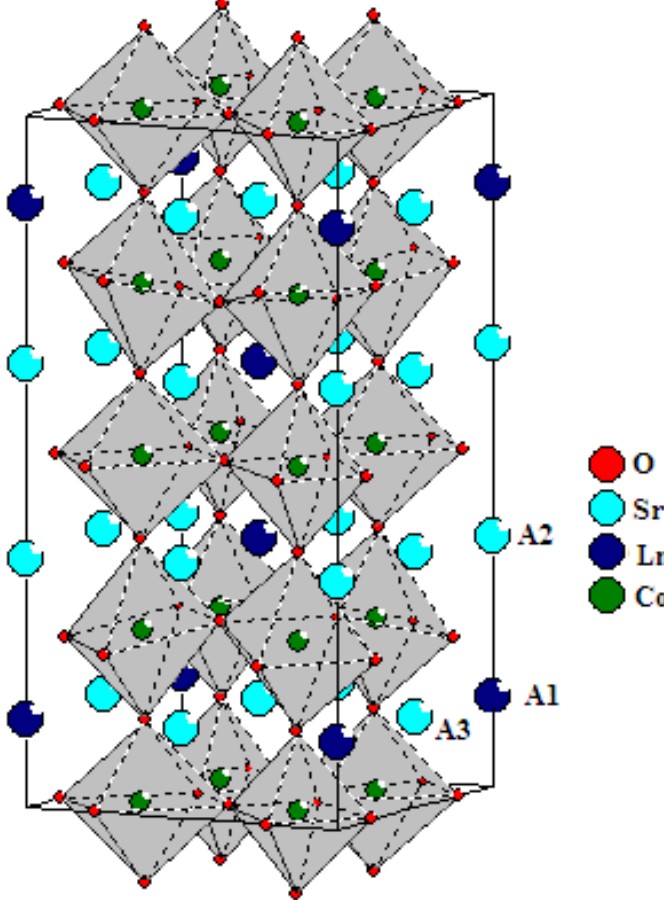

**Figure 1.** Structural model of the unit cell for a tetragonal $Ln_{0.25}Sr_{0.75}CoO_{3-\delta}$ complex oxide.

The crystal structure of $Gd_{1-x}Sr_xCoO_{3-\delta}$ oxide is significantly influenced by heat treatment conditions [29,30,32–36]. At temperatures above 1473 K, the oxide with $x = 0.8$ possesses a cubic perovskite structure with a statistical distribution of strontium and

gadolinium ions in the A-sublattice. Slow cooling of the sample to 1363 K leads to the ordering of Gd/Sr cations in the A-sublattice and changes the structure to tetragonal [32,33]. The transition temperature from a tetragonal ordered to a cubic disordered structure for the oxide with $x = 0.9$ was defined as 1263 K [35].

The oxygen content in the $Ln_{1-x}Sr_xCoO_{3-\delta}$ (Ln = Sm, Gd, Dy, Y, Ho) oxides slightly decreases with decreasing lanthanide radius and much more significantly with increasing strontium content [26,27,29,30,37,38].

The main disadvantage of Sr-substituted rare earth cobaltites in practical application is their weak thermal and $Po_2$-stability [22,39,40]. It is generally accepted that the partial substitution of Co ions by Fe improves the stability of oxides without significantly deteriorating important functional properties.

This work focuses on the influence of Fe-substitution degree on the homogeneity range, crystal structure, oxygen nonstoichiometry, and electrical properties of Sr-enriched $Gd_{1-x}Sr_xCo_{1-y}Fe_yO_{3-\delta}$ with $x = 0.8$ and 0.9.

## 2. Results and Discussion

### 2.1. Crystal Structure of $Gd_{1-x}Sr_xCo_{1-y}Fe_yO_{3-\delta}$

In order to determine the homogeneity range of iron substituted gadolinium strontium cobaltate, a series of samples with the overall composition $Gd_{1-x}Sr_xCo_{1-y}Fe_yO_{3-\delta}$, where $x = 0.8$; 0.9 and $0.1 \leq y \leq 0.9$ in a step of $y = 0.1$, was prepared in air by the glycerol-nitrate technique. According to the results of XRD analysis all samples were identified as single-phase. As an example, Figure 2 demonstrates XRD patterns for the selected $Gd_{0.2}Sr_{0.8}Co_{1-y}Fe_yO_{3-\delta}$ oxides.

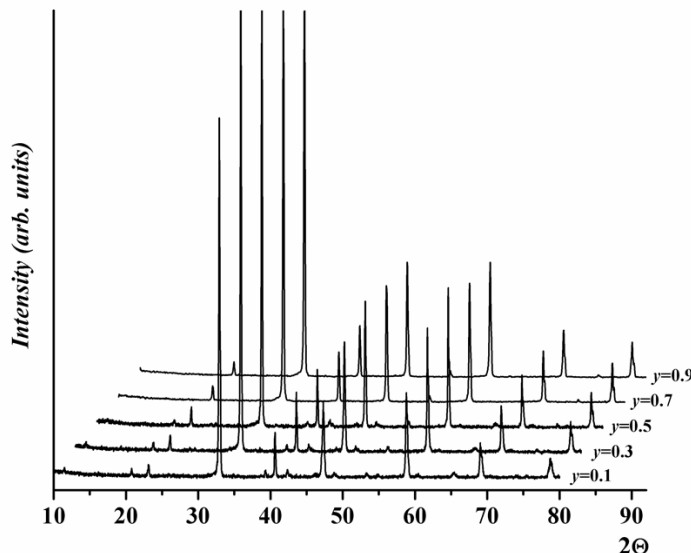

**Figure 2.** XRD patterns of the $Gd_{0.2}Sr_{0.8}Co_{1-y}Fe_yO_{3-\delta}$ solid solutions with various iron content.

The crystal structure of quenched $Gd_{0.2}Sr_{0.8}Co_{1-y}Fe_yO_{3-\delta}$ oxides with $0.1 \leq y \leq 0.5$ is similar to the Fe-undoped cobaltate $Gd_{0.2}Sr_{0.8}CoO_{3-\delta}$ [30]. The XRD patterns for all of them contain weak diffraction peaks in vicinity of $2\theta \approx 21°$ and $2\theta \approx 39°$ (corresponding to the interlayered distances $d \approx 4.26$ (hkl = 103) and $d \approx 2.29$ (hkl = 215)) that could be attributed to the tetragonal superstructure [27–31]. The latter is formed due to the ordered location of Ln and Sr cations in the A-sublattice of perovskite structure and the ordering of the oxygen vacancies. Thus, the introduction of iron into the cobalt sublattice in $Gd_{0.2}Sr_{0.8}CoO_{3-\delta}$ to some extent (up to $y = 0.5$) does not disturb the tetragonal superstructure, i.e., Sr and Gd ordering in the A-sublattice.

The XRD patterns of single-phase samples $Gd_{0.2}Sr_{0.8}Co_{1-y}Fe_yO_{3-\delta}$ with $0.1 \leq y \leq 0.5$ were refined by the Rietveld method within the tetragonal structure $2a_p \times 2a_p \times 4a_p$ (SG

$I4/mmm$). The typical Rietveld refinement profiles for $Gd_{0.2}Sr_{0.8}Co_{1-y}Fe_yO_{3-\delta}$ with $y = 0.1$ and 0.5 are shown in Figure 3, as an example. The values of the refined unit cell parameters and atom coordinates for the $Gd_{0.2}Sr_{0.8}Co_{1-y}Fe_yO_{3-\delta}$ ($0.1 \leq y \leq 0.5$) solid solutions are presented in Tables 1 and 2, respectively.

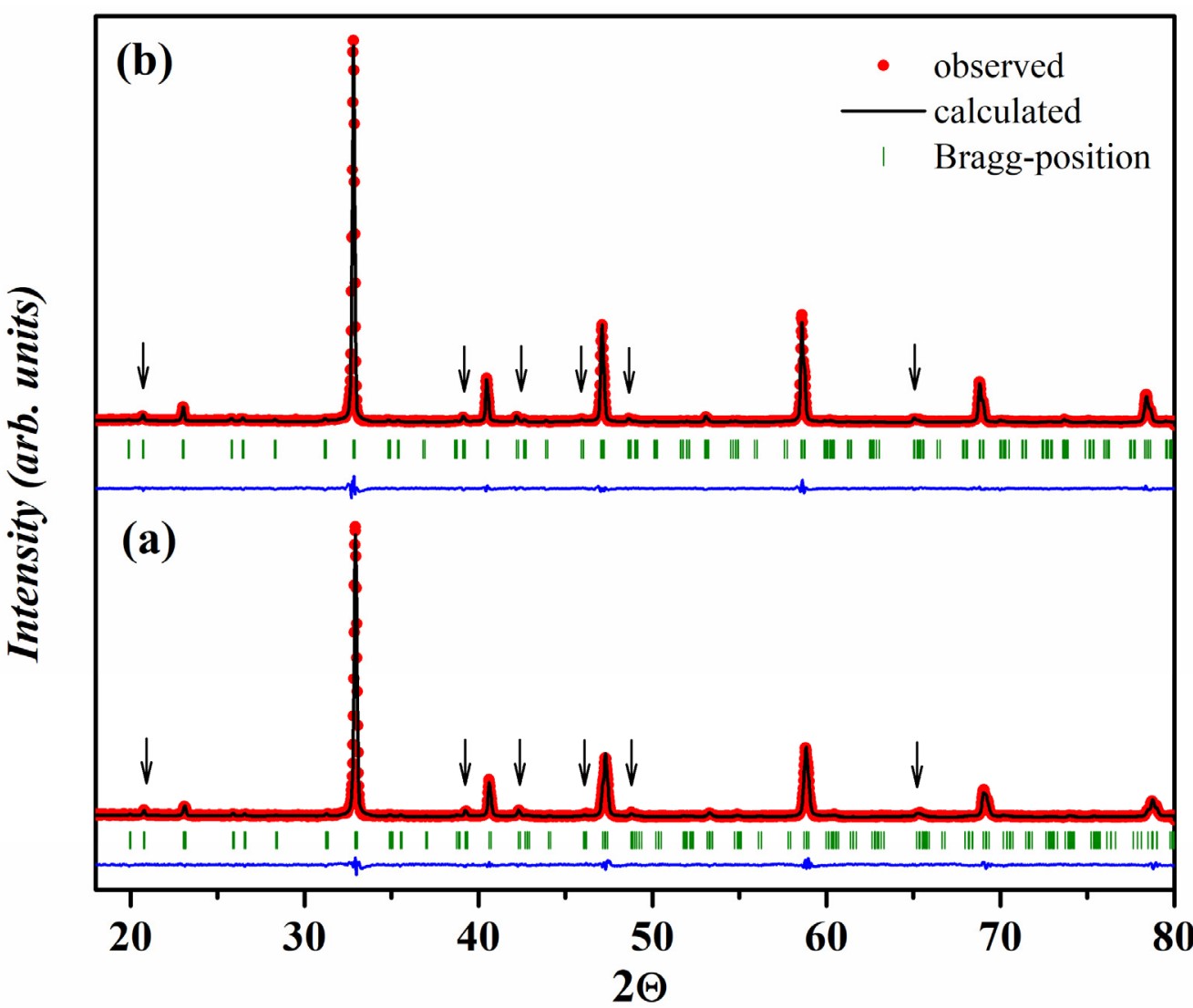

**Figure 3.** XRD patterns for $Gd_{0.2}Sr_{0.8}Co_{1-y}Fe_yO_{3-\delta}$: $y = 0.1$ (**a**) and $y = 0.5$ (**b**) refined by the Rietveld method. Points represent the experimental data and the solid curve is the calculated profile. A difference curve is plotted at the bottom. Vertical lines mark the positions of allowed Bragg reflections. Arrows indicate super-structural reflections for the $2a_p \times 2a_p \times 4a_p$ tetragonal cell.

**Table 1.** The unit cell parameters (SG $I4/mmm$) of $Gd_{0.2}Sr_{0.8}Co_{1-y}Fe_yO_{3-\delta}$ ($0.1 \leq y \leq 0.5$) quenched from 1373 K in air and *R*-factors refined by the Rietveld method.

| $y$ | $a$, Å | $c$, Å | $V$, (Å)$^3$ | R-Factors, % | | |
|---|---|---|---|---|---|---|
| | | | | $R_{Br}$ | $R_f$ | $R_p$ |
| 0.1 | 7.674(1) | 15.401(1) | 907.15(2) | 6.32 | 11.3 | 8.14 |
| 0.2 | 7.685(1) | 15.403(1) | 909.79(2) | 5.61 | 9.66 | 7.53 |
| 0.3 | 7.693(1) | 15.408(1) | 912.13(1) | 5.85 | 11.6 | 9.67 |
| 0.4 | 7.697(1) | 15.412(1) | 913.09(3) | 6.24 | 11.5 | 7.19 |
| 0.5 | 7.704(1) | 15.426(1) | 915.59(3) | 6.13 | 9.25 | 7.82 |

**Table 2.** Atomic coordinates in the tetragonal cell of $Gd_{0.2}Sr_{0.8}Co_{0.8}Fe_{0.2}O_{3-\delta}$ refined by the Rietveld analysis.

| Atom | $x$ | $y$ | $z$ |
|:---:|:---:|:---:|:---:|
| Co1/Fe1 | 0.245(1) | 0.245(1) | 0 |
| Co2/Fe2 | 0.250 | 0.250 | 0.250 |
| Sr1/Gd1 | 0 | 0 | 0.138(1) |
| Sr2 | 0 | 0 | 0.627(1) |
| Sr3 | 0 | 0.5 | 0.130(1) |
| O1 | 0.221(2) | 0.221(2) | 0.117(1) |
| O2 | 0.179(1) | 0 | 0 |
| O3 | 0.211(3) | 0.5 | 0 |
| O4 | 0 | 0.238(4) | 0.251(1) |

Further introduction of iron into the cobalt sublattice at $x = 0.8$ or a decrease in the gadolinium content, regardless of the Co/Fe ratio, leads to a change in the structure from tetragonal with an ordered location of $Gd^{3+}$ and $Sr^{2+}$ in the A-sublattice to cubic with a statistical distribution of cations. The XRD patterns of $Gd_{1-x}Sr_xCo_{1-y}Fe_yO_{3-\delta}$ with $x = 0.8$, $0.6 \leq y \leq 0.9$ and $x = 0.9$, $0.1 \leq y \leq 0.9$ were refined in the ideal cubic perovskite structure (SG *Pm-3m*), similar to a Co-free gadolinium strontium ferrite $Gd_{1-x}Sr_xFeO_{3-\delta}$ [36,37]. This result correlates well with the fact that $Sr_{0.9}Gd_{0.1}CoO_{3-\delta}$ quenched from 1373 K in air possesses a disordered cubic perovskite structure, and the structural transition from tetragonal to cubic structure occurs at 1263 K [29,30,32,35].

The XRD patterns for $Gd_{1-x}Sr_xCo_{1-y}Fe_yO_{3-\delta}$ with $x = 0.8$, $y = 0.7$ and $x = 0.9$, $y = 0.3$ processed by the Rietveld method are shown in Figure 4 as an example. The refined structural parameters for all single-phase $Gd_{1-x}Sr_xCo_{1-y}Fe_yO_{3-\delta}$ with the cubic structure are listed in Table 3.

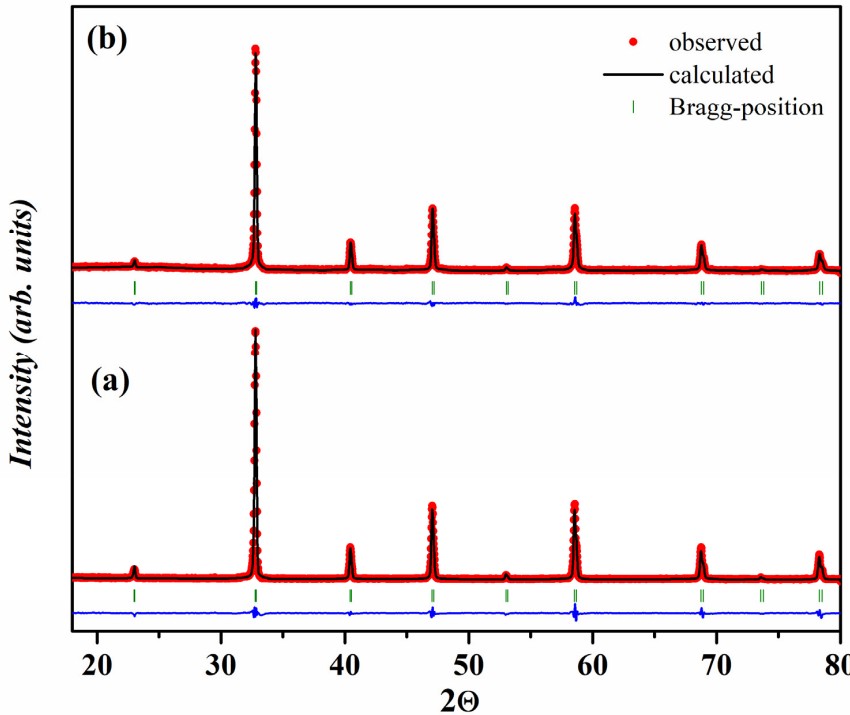

**Figure 4.** XRD patterns for $Gd_{1-x}Sr_xCo_{1-y}Fe_yO_{3-\delta}$: (**a**) $x = 0.8$, $y = 0.7$ and (**b**) $x = 0.9$, $y = 0.3$ refined by the Rietveld method. Points represent the experimental data and the solid curve is the calculated profile. A difference curve is plotted at the bottom. Vertical lines represent the positions of allowed Bragg reflections.

**Table 3.** The structural parameters for the $Gd_{1-x}Sr_xCo_{1-y}Fe_yO_{3-\delta}$ solid solution quenched from 1373 K in air (SG *Pm-3m*) and *R*-factors refined by the Rietveld method.

| | | SG *Pm-3m*: Gd/Sr (0.5;0.5;0.5); Fe/Co (0; 0; 0); O (0.5; 0; 0) | | | | | R-Factors, % | | |
|---|---|---|---|---|---|---|---|---|---|
| *x* | *y* | *a*, Å | *V*, (Å)³ | $d_{Fe/Co-O}$, Å | $d_{Gd/Sr-O}$, Å | $d_{Gd/Sr-Fe/Co}$, Å | $R_{Br}$ | $R_f$ | $R_p$ |
| | 0.6 | 3.851(1) | 57.11(1) | 1.925(1) | 2.723(1) | 3.335(1) | 5.06 | 4.84 | 8.46 |
| | 0.7 | 3.857(1) | 57.40(1) | 1.928(1) | 2.727(1) | 3.340(1) | 3.79 | 4.85 | 7.04 |
| 0.8 | 0.8 | 3.860(1) | 57.52(1) | 1.930(1) | 2.729(1) | 3.343(1) | 5.35 | 4.19 | 11.6 |
| | 0.9 | 3.864(1) | 57.72(1) | 1.932(1) | 2.732(1) | 3.346(1) | 4.52 | 3.93 | 9.60 |
| | 0.1 | 3.849(1) | 57.03(1) | 1.924(1) | 2.721(1) | 3.333(1) | 5.46 | 4.24 | 9.58 |
| | 0.3 | 3.856(1) | 57.36(1) | 1.928(1) | 2.727(1) | 3.340(1) | 2.49 | 2.21 | 7.24 |
| 0.9 | 0.5 | 3.861(1) | 57.59(1) | 1.930(1) | 2.730(1) | 3.344(1) | 5.95 | 3.93 | 7.88 |
| | 0.7 | 3.864(1) | 57.70(1) | 1.932(1) | 2.732(1) | 3.346(1) | 3.33 | 2.41 | 9.59 |
| | 0.9 | 3.870(1) | 57.99(1) | 1.934(1) | 2.734(1) | 3.348(1) | 1.11 | 2.98 | 7.90 |

The unit cell parameters and unit cell volume of $Gd_{1-x}Sr_xCo_{1-y}Fe_yO_{3-\delta}$ increase linearly with increasing iron content both in tetragonal (Figure 5a) and cubic (Figure 5b) structures due to the relative cation sizes of cations, since the ionic radius of iron ($r_{Fe}^{3+}$(HS) = 0.645 Å, CN = 6) is greater than that of cobalt ($r_{Co}^{3+}$(LS) = 0.545 Å, $r_{Co}^{3+}$(HS) = 0.61 Å, CN = 6) [41].

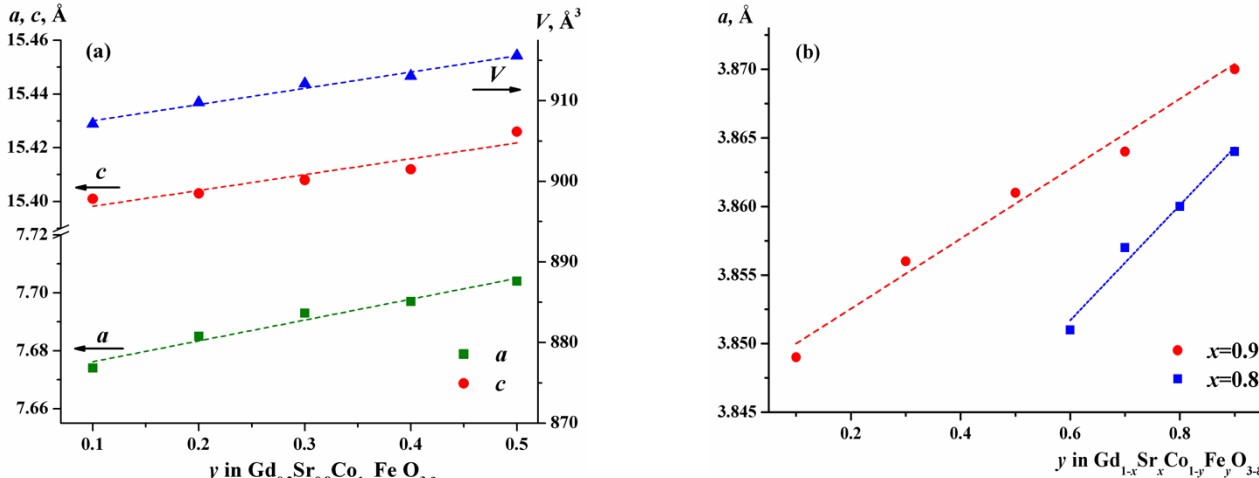

**Figure 5.** The unit cell parameters of $Gd_{1-x}Sr_xCo_{1-y}Fe_yO_{3-\delta}$ versus iron content (*y*): (**a**) *x* = 0.8, $0.1 \leq y \leq 0.5$; (**b**) *x* = 0.8, $0.6 \leq y \leq 0.9$ and *x* = 0.9, $0.1 \leq y \leq 0.9$.

To compare the unit cell parameters of $Gd_{0.2}Sr_{0.8}Co_{1-y}Fe_yO_{3-\delta}$ over the entire range of iron concentrations ($0.1 \leq y \leq 0.9$), the tetragonal supercell parameters were recalculated to the pseudo-cubic cell ($a_{cub}$) using the formula:

$$a_{cub} = (V/z)^{1/3}, \tag{1}$$

where *V* is the volume of a tetragonal supercell and *z* is the number of $ABO_3$ formula units belonging to one supercell (*z* = 16 for $2a_p \times 2a_p \times 4a_p$ supercell) (Figure 6). One can see that the plot of pseudo-cubic cell parameter ($a_{cub}$) versus iron content in $Gd_{0.2}Sr_{0.8}Co_{1-y}Fe_yO_{3-\delta}$ shows a visible gap between *y* = 0.5 and *y* = 0.6. This concentration range corresponds to an "order-disorder" structural transition similar to that observed for $Gd_{1-x}Sr_xCoO_{3-\delta}$ with increasing Sr content [29,30].

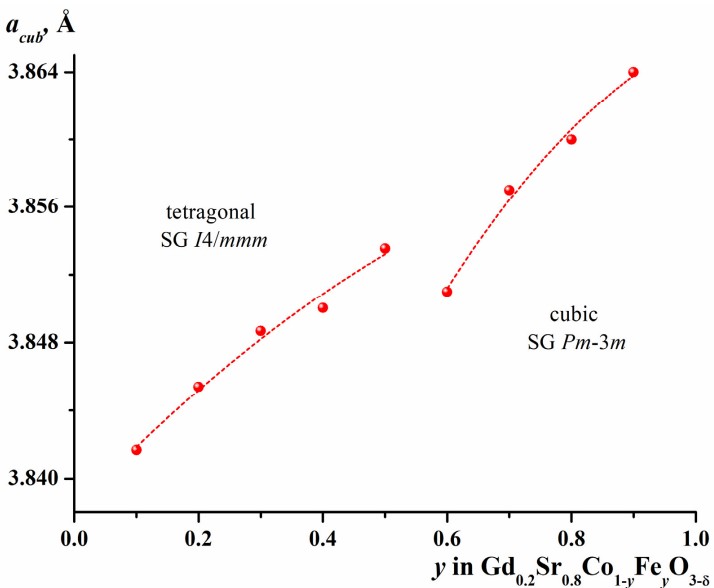

**Figure 6.** The pseudo-cubic cell parameters of the quenched $Gd_{0.2}Sr_{0.8}Co_{1-y}Fe_yO_{3-\delta}$ oxides versus iron content ($y$).

### 2.2. Oxygen Content of $Gd_{1-x}Sr_xCo_{1-y}Fe_yO_{3-\delta}$

Figure 7 demonstrates the changes in oxygen content for $Gd_{1-x}Sr_xCo_{1-y}Fe_yO_{3-\delta}$ with $x = 0.8$, $0.1 \leq y \leq 0.9$ in step of $y = 0.2$ and $x = 0.9$, $y = 0.7$ versus temperature in air measured by TGA. The absolute value of oxygen content at room temperature and at 1373 K are listed in Table 4. As expected, the incorporation of iron into the cobalt sublattice $Gd_{0.2}Sr_{0.8}Co_{1-y}Fe_yO_{3-\delta}$ leads to a gradual increase in oxygen content. Since iron is a more electropositive element compared to cobalt ($\chi_{Fe} = 1.64$ and $\chi_{Co} = 1.7$ [42]), it acts as an electron donor ($Fe_{Co}^{\bullet}$), preventing formation of oxygen vacancies ($V_O^{\bullet\bullet}$). On the other hand, the substitution of strontium for gadolinium in $Gd_{1-x}Sr_xCo_{0.3}Fe_{0.7}O_{3-\delta}$ decreases the oxygen content, since strontium serves as an acceptor-type defect ($Sr'_{Gd}$) and therefore facilitates oxygen release. An excessive negative charge of acceptor-type defect ($Sr'_{Gd}$) is compensated by both oxygen vacancies ($V_O^{\bullet\bullet}$) and electron holes localized on $3d$-transition metal ions.

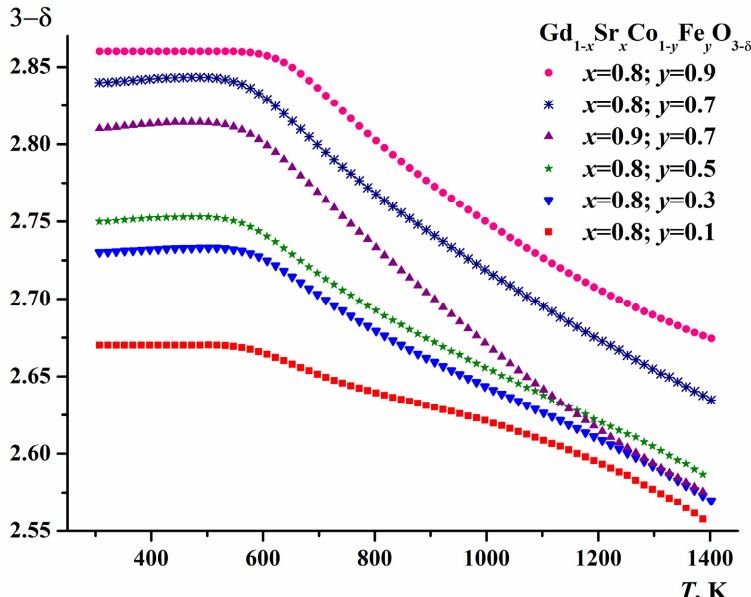

**Figure 7.** Temperature dependencies of oxygen content in $Gd_{1-x}Sr_xCo_{1-y}Fe_yO_{3-\delta}$ in air.

**Table 4.** Oxygen content and mean oxidation state of 3*d*-transition metals in $Gd_{1-x}Sr_xCo_{1-y}Fe_yO_{3-\delta}$ in air.

| Composition | *T*, K | Oxygen Content 3-$\delta$ | Mean Oxidation State of 3*d*-Transition Metals |
|---|---|---|---|
| $Gd_{0.2}Sr_{0.8}Co_{0.9}Fe_{0.1}O_{3-\delta}$ | 298 | 2.67 ± 0.01 | 3.14 |
| | 1373 | 2.56 ± 0.01 | 2.92 |
| $Gd_{0.2}Sr_{0.8}Co_{0.7}Fe_{0.3}O_{3-\delta}$ | 298 | 2.73 ± 0.01 | 3.26 |
| | 1373 | 2.57 ± 0.01 | 2.94 |
| $Gd_{0.2}Sr_{0.8}Co_{0.5}Fe_{0.5}O_{3-\delta}$ | 298 | 2.75 ± 0.01 | 3.30 |
| | 1373 | 2.59 ± 0.01 | 2.98 |
| $Gd_{0.2}Sr_{0.8}Co_{0.3}Fe_{0.7}O_{3-\delta}$ | 298 | 2.84 ± 0.01 | 3.48 |
| | 1373 | 2.64 ± 0.01 | 3.08 |
| $Gd_{0.2}Sr_{0.8}Co_{0.1}Fe_{0.9}O_{3-\delta}$ | 298 | 2.86 ± 0.01 | 3.52 |
| | 1373 | 2.67 ± 0.01 | 3.14 |
| $Gd_{0.1}Sr_{0.9}Co_{0.3}Fe_{0.7}O_{3-\delta}$ | 298 | 2.81 ± 0.01 | 3.52 |
| | 1373 | 2.58 ± 0.01 | 3.06 |

The dependences of the oxygen content in $Gd_{0.2}Sr_{0.8}Co_{1-y}Fe_yO_{3-\delta}$ versus iron concentration at room temperature and at 1373 K show a bend between $y = 0.5$ and $y = 0.6$ which also confirms the structural transition similar to the pseudo-cubic cell parameter (Figure 8).

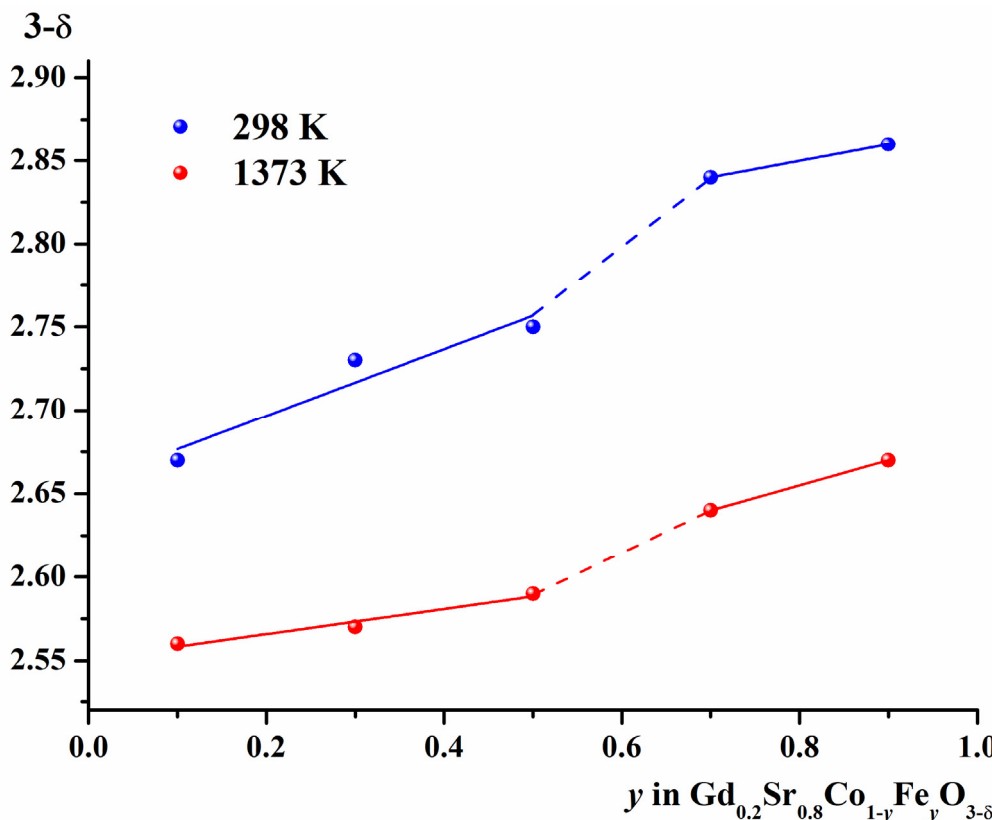

**Figure 8.** The oxygen content in $Gd_{0.2}Sr_{0.8}Co_{1-y}Fe_yO_{3-\delta}$ versus iron content.

The total variation in oxygen content for each oxide within the temperature range 298–1373 K for the tetragonal $Gd_{0.2}Sr_{0.8}Co_{1-y}Fe_yO_{3-\delta}$ ($y = 0.1–0.5$) is obviously lower ($\Delta\delta = 0.11–0.16$) compared to that observed for oxides ($y = 0.7–0.9$) with the cubic structure ($\Delta\delta = 0.19–0.20$). This also supports the existence of a "order-disorder" transition since the tetragonal oxides were characterized not only by a cationic superstructure, but also

with the oxygen vacancies ordering [26,27,31]. As a rule, the variation in oxygen content for phases ordered in anionic sublattice is always smaller compared to that for the related disordered oxides.

Since the oxidation states of strontium, gadolinium, and oxygen ions are considered unchanged, unlike 3$d$-transition metals (Co, Fe), the formula of solid solution can be written as $Gd_{1-x}^{3+}Sr_x^{2+}Me^{z+}O_{3-\delta}$ (Me = Co and Fe), where $z$ is mean oxidation state of 3$d$-transition metal ions. Thus, the electroneutrality equation can be written as follows:

$$3 \cdot (1-x) + 2 \cdot x + z = 2 \cdot (3 - \delta) \text{ or } z = 2 \cdot (3 - \delta) + x - 3 \tag{2}$$

The mean oxidation state of 3$d$-transition metal ions ($z$) at room temperature and at 1373 K calculated by Equation (3) (see Table 4) increases with increasing iron content. Considering that Co is a more electronegative cation compared to Fe, it can be assumed that at a mean oxidation state value above 3+, iron ions will be the first to increase their oxidation state to a $Fe^{4+}$ form. On the other hand, in the case of the mean oxidation state of 3$d$-transition metal ions being lower than 3+, parts of cobalt ions will be transformed to a $Co^{2+}$ form. According to the electroneutrality condition and the values of oxygen content, the formulas of the $Gd_{0.2}Sr_{0.8}Co_{1-y}Fe_yO_{3-\delta}$ solid solution can be presented as follows:

at 298 K:

$Gd_{0.2}Sr_{0.8}Co_{0.86}^{3+}Co_{0.04}^{4+}Fe_{0.1}^{4+}O_{2.67}$

$Gd_{0.2}Sr_{0.8}Co_{0.7}^{3+}Fe_{0.04}^{3+}Fe_{0.26}^{4+}O_{2.73}$

$Gd_{0.2}Sr_{0.8}Co_{0.5}^{3+}Fe_{0.2}^{3+}Fe_{0.3}^{4+}O_{2.75}$

$Gd_{0.2}Sr_{0.8}Co_{0.3}^{3+}Fe_{0.22}^{3+}Fe_{0.48}^{4+}O_{2.84}$

$Gd_{0.2}Sr_{0.8}Co_{0.1}^{3+}Fe_{0.38}^{3+}Fe_{0.52}^{4+}O_{2.86}$

at 1373 K:

$Gd_{0.2}Sr_{0.8}Co_{0.82}^{3+}Co_{0.08}^{2+}Fe_{0.1}^{3+}O_{2.56}$

$Gd_{0.2}Sr_{0.8}Co_{0.64}^{3+}Co_{0.06}^{2+}Fe_{0.3}^{3+}O_{2.57}$

$Gd_{0.2}Sr_{0.8}Co_{0.48}^{3+}Co_{0.02}^{2+}Fe_{0.5}^{3+}O_{2.59}$

$Gd_{0.2}Sr_{0.8}Co_{0.3}^{3+}Fe_{0.62}^{3+}Fe_{0.08}^{4+}O_{2.64}$

$Gd_{0.2}Sr_{0.8}Co_{0.1}^{3+}Fe_{0.76}^{3+}Fe_{0.14}^{4+}O_{2.67}$

The obtained results illustrate that the introduction of iron in $Gd_{0.2}Sr_{0.8}Co_{1-y}Fe_yO_{3-\delta}$ significantly changes the ratio of 3$d$-transition metals in various oxidation states and increases the amount of $Fe^{4+}$ ions at least at room temperature in air. An increase in temperature leads to a decrease in the mean oxidation state of 3$d$-metal ions.

### 2.3. Electrical Conductivity of $Gd_{1-x}Sr_xCo_{1-y}Fe_yO_{3-\delta}$

The temperature dependences of the Seebeck coefficient (Q) and total electrical conductivity (σ) for $Gd_{0.2}Sr_{0.8}Co_{0.3}Fe_{0.7}O_{3-\delta}$ together with Fe-free cobaltite and Co-free ferrite taken from [30,43] are shown in Figure 9a,b, correspondingly.

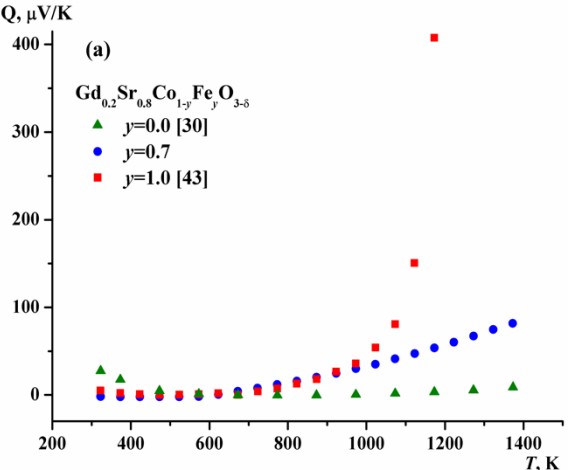
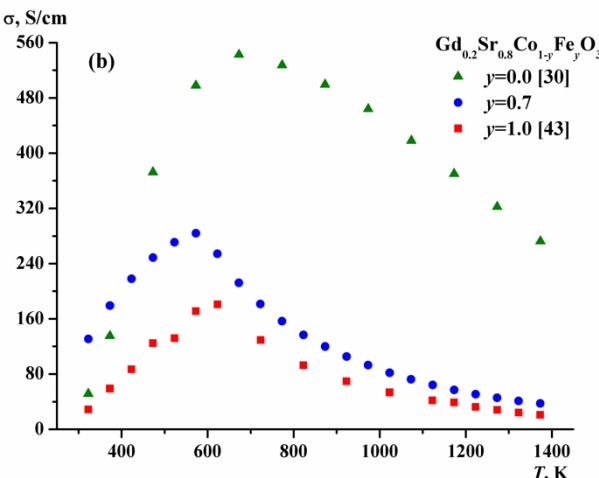

**Figure 9.** Temperature dependencies of the Seebeck coefficient (**a**) and total electrical conductivity (**b**) for $Gd_{0.2}Sr_{0.8}Co_{1-y}Fe_yO_{3-\delta}$ ($y$ = 0.0; 0.7; 1.0) in air.

Positive values of Seebeck coefficient in $Gd_{0.2}Sr_{0.8}Co_{0.3}Fe_{0.7}O_{3-\delta}$ within the entire temperature range (298–1373 K) indicate a predominant $p$-type conductivity, which is in

good correlation with the results obtained for the related rare earth/strontium cobaltites and ferrites [20,21,23,24,30,38,43,44].

The shape of temperature dependence for total conductivity in mixed $Gd_{0.2}Sr_{0.8}Co_{0.3}Fe_{0.7}O_{3-\delta}$ is generally the same as for the parent $Gd_{0.2}Sr_{0.8}MeO_{3-\delta}$ (Me = Fe, Co) oxides [30,43]. All of them possess a maximum at approximately 600 K. The room temperature value for $Gd_{0.2}Sr_{0.8}Co_{0.3}Fe_{0.7}O_{3-\delta}$ is noticeably higher than that for parent cobaltite and ferrite since the former has high concentration of charge carriers $Gd_{0.2}Sr_{0.8}Co_{0.1}^{3+}Fe_{0.38}^{3+}Fe_{0.52}^{4+}O_{2.86}$, i.e., localized holes, already at room temperature. At relatively low temperatures ($T < 600$ K), where oxygen exchange between the solid and gas phases is negligible, the rise of total electrical conductivity is caused by increasing the mobility of holes localized on $3d$-transition metals ($Me^{\bullet}$) and partly by the increasing of charge carrier concentration due to the disproportionation reaction:

$$2Me^{\times} = Me^{\bullet} + Me^{/} \tag{3}$$

This disproportionation reaction (3) occurs more easily for $Me$ = Co rather than for $Me$ = Fe. This could be a reason for the more rapid increase of conductivity for Fe-free cobaltite compared to the mixed Co/Fe oxide and even more strongly compared to Co-free ferrite (Figure 9b). A decrease in conductivity with increasing Fe content in mixed Co/Fe oxides is often explained by considering ($Fe^{\bullet}$)as a "trap" for the holes (main charge carriers).

At $T > 600$ K, a decrease of total conductivity is associated with the active release of oxygen from the crystal lattice, which results in the formation of oxygen vacancies ($V_O^{\bullet\bullet}$) and a decrease in the concentration of main charge carriers—holes, according to the reaction:

$$O_O^{\times} + 2Me^{\bullet} = \frac{1}{2}O_2 + V_O^{\bullet\bullet} + 2Me^{\times} \tag{4}$$

where $Me$ denotes Fe or Co atoms. The formation of oxygen vacancies disrupts the transport pathways $Me$–O–$Me$ for the charge carriers which also impairs conductivity.

The activation energy for conductivity in $Gd_{0.2}Sr_{0.8}Co_{0.3}Fe_{0.7}O_{3-\delta}$, calculated from the slope of the $\lg(\sigma \cdot T) = f(1/T)$ dependency in the temperature range 300–700 K, is equal 0.17 eV, which is typical for the small radius polaron-type mechanism.

Changes in the value of the Seebeck coefficient with temperature for the oxide with $y = 0.7$ measured in our work are in between the results reported earlier [30,43] for the unsubstituted cobaltite and ferrite and are in good agreement with the proposed conductivity model. At $T < 600$ the conductivity increases mainly due to a raise in the mobility of charge carriers while their concentration remains unchanged. Thus, according to the Heikes formula [45], the value of the Seebeck coefficient should decrease or remain unchanged. At $T > 600$ K, the concentration of charge carriers decreases with temperature (Equation (4)) and the Heikes formula [45] predicts an increase in the Seebeck coefficient. It is worth noting that the charge disproportionation process will also affect the concentration of $3d$ metals in various oxidation states, so a more detailed discussion of this issue at present is not possible.

## 3. Materials and Methods

Polycrystalline samples of $Gd_{1-x}Sr_xCo_{1-y}Fe_yO_{3-\delta}$ ($x = 0.8$; 0.9 and $0.1 \leq y \leq 0.9$) were prepared by a glycerol-nitrate technique from stoichiometric amounts of pre-calcined gadolinium oxide $Gd_2O_3$ (99.99% purity, Lanhit, Russia), strontium carbonate $SrCO_3$ (99.99% purity, Lanhit, Russia), iron oxalate dihydrate $FeC_2O_4 \cdot 2H_2O$ (99.0% purity, Ormet, Russia) and metallic cobalt. Metallic cobalt was prepared by the reduction of cobalt oxide $Co_3O_4$ (99.0% purity, Ormet, Russia) at 673–873 K in a hydrogen flow during 6 h. Further details of glycerol-nitrate technique were described elsewhere [24,30]. Final anneals of the $Gd_{1-x}Sr_xCo_{1-y}Fe_yO_{3-\delta}$ samples were performed at 1373 K in air for 80 h with intermediate grinding. All samples were quenched to room temperature by removing from a furnace to a cold massive metallic plate (cooling rate of about 500 K/min).

The synthesized oxides were characterized by X-ray diffraction (XRD) analysis using a Shimadzu XRD 7000 instrument, Japan ($Cu_{K\alpha}$–radiation, angle range $2\Theta = 10°–80°$, scan

step 0.02 with the exposure time 5 s). The structural parameters were refined by the Rietveld profile method using the Fullprof-2011 software.

Thermogravimetric measurements (TGA) were carried out using a simultaneous thermal analyzer, STA 409 PC Luxx, Netzsch, Germany, within the temperature range $298 \leq T, K \leq 1373$ in air. The measurements were performed with the cooling/heating rate equal to 2 K/min. The absolute values of oxygen content in the sample were determined using a direct reduction of the complex oxides in TG cell by hydrogen (10% $N_2$–90% $H_2$) at 1423 K according to the following reaction:

$$Gd_{1-x}Sr_xCo_{1-y}Fe_yO_{3-\delta} + \tfrac{1}{2}(3 - 2\delta + x)H_2 \rightarrow \tfrac{1-x}{2}Gd_2O_3 + xSrO + (1-y)Co + yFe + \tfrac{1}{2}(3 - 2\delta + x)H_2O \quad (5)$$

The phase composition of the samples after reduction was confirmed by XRD analysis. The accuracy of the oxygen index determination was not less than $\pm 0.01$.

Total electrical conductivity ($\sigma$) was studied using a 4-probe method in the temperature range $298 \leq T, K \leq 1373$ in air. The sample for measurements was preliminarily compacted into the form of a bar (4 mm $\times$ 4 mm $\times$ 25 mm) and sintered at 1573 K in air for 24 h, with subsequent slow cooling (1 K/min) to room temperature. The density of the polished ceramic sample was 92% of its theoretical value calculated from the XRD data. The Seebeck coefficient was measured simultaneously with conductivity using two Pt-Pt/Rh thermocouples at both ends of the ceramic sample.

## 4. Conclusions

It is shown that the crystal structure of $Gd_{1-x}Sr_xCo_{1-y}Fe_yO_{3-\delta}$ complex oxides significantly depends on strontium and iron contents. Fe-poor $Gd_{1-x}Sr_xCo_{1-y}Fe_yO_{3-\delta}$ oxides with $x = 0.8$, $0.1 \leq y \leq 0.5$ possess a tetragonal $2a_p \times 2a_p \times 4a_p$ superstructure (SG $I4/mmm$) with the ordered location of $Gd^{3+}$ and $Sr^{2+}$ cations in the A-sublattice, whereas oxides with $x = 0.8$, $0.6 \leq y \leq 0.9$ and $x = 0.9$, $0.1 \leq y \leq 0.9$ possess a cubic structure (SG $Pm$-$3m$) with the statistically distributed cations in the A-sublattice. The unit cell parameters and unit cell volume are gradually increased with increasing iron content. It is shown that the oxygen content decreases with an increase in temperature and an increase in the concentration of strontium and cobalt in the samples. The change in the oxygen content for oxides with the tetragonal layered superstructure is visibly lower than that for oxides with the cubic disordered perovskite structure. The positive value of the Seebeck coefficient confirmed $p$-type conductivity. Total conductivity of mixed $Gd_{0.2}Sr_{0.8}Co_{0.3}Fe_{0.7}O_{3-\delta}$ oxide at room temperature exceeds the values for "pure" ferrite and cobaltite. However, at $T > 400$ K, an increase in the Fe content in $Gd_{0.2}Sr_{0.8}Co_{1-y}Fe_yO_{3-\delta}$ results in a decrease of conductivity value.

**Author Contributions:** Conceptualization, V.A.C. and T.V.A.; methodology, V.A.C. and T.V.A.; software, T.V.A.; validation, D.K.M. and T.V.A.; formal analysis, D.K.M. and T.V.A.; investigation, D.K.M. and T.V.A.; resources, V.A.C.; data curation, V.A.C. and T.V.A.; writing—original draft preparation, T.V.A.; writing—review and editing, V.A.C.; visualization, V.A.C. and T.V.A.; supervision, V.A.C.; project administration, V.A.C.; funding acquisition, V.A.C. All authors have read and agreed to the published version of the manuscript.

**Funding:** Ministry of Science and Higher Education of Russian Federation (State Task number AAAA-A20-120061990010-7).

**Acknowledgments:** The work was financially supported by the Ministry of Science and Higher Education of Russian Federation (State Task number AAAA-A20-120061990010-7).

**Conflicts of Interest:** The authors declare no conflict of interest.

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
