# Peer review of "Crystal Structure and Properties of Gd1-xSrxCo1-yFeyO3-δ Oxides as Promising Materials for Catalytic and SOFC Application"

_catalysts, doi:10.3390/catal12111344_

Round 1

Reviewer 1 Report

Paper presents interesting results clearly related to solid state chemistry and solid state ionics including SOFC. If Catalysts Editor is ready to publish this paper not related directly to catalysis, it can be accepted. In this case authors are advised to add comments what is the meaning of their research for catalysis. English is to be slightly corrected. If Editor  will decide to transfer this article to more suitable MDPI Journals (Energies, etc) , it will be his choice. 

Author Response

The authors would like to thank the Reviewers for their valuable comments that aimed to improve the manuscript at the peer-review stage. All Reviewers’ comments have been considered and required corrections were made. All corrections in the revised manuscript are marked with yellow highlighting.

Answer: We believe that the information provided in the manuscript, i.e. crystal structure, oxygen content, mean oxidation state of 3d-transition metals and possibly electrical properties (conductivity and the Seebeck coefficient) are important in selecting an appropriate composition that can be used as catalyst. It is widely acknowledged that the variable oxidation state of Co and Fe ions and the concentration of oxygen vacancies play an important role in the catalytic activity of materials that are considered as catalysts for various Red-Ox processes. Additional comment on this matter was inserted into the Introduction.

The English was additionally checked, and the necessary corrections were made.

Reviewer 2 Report

The reviewed papers contains new interesting results and may be published in "Catalysts" after minor correction.

Page 3, lines 87-88. Authors write that "...details of glycerol-nitrate technique were described elsewhere [24,31]...". But in reference of [31] samples were prepared not by authors of reviewed paper and sinthesys was performed using the solid-state reactions method. It seems that right reference is not [31] but [30].

It would be interested if authors can give more detailed description and commentary of the results of study of Seebeck coefficient of the sample.

Author Response

The authors would like to thank the Reviewers for their valuable comments that aimed to improve the manuscript at the peer-review stage. All Reviewers’ comments have been considered and required corrections were made. All corrections in the revised manuscript are marked with yellow highlighting.

1) Page 3, lines 87-88. Authors write that "...details of glycerol-nitrate technique were described elsewhere [24,31]...". But in reference of [31] samples were prepared not by authors of reviewed paper and sinthesys was performed using the solid-state reactions method. It seems that right reference is not [31] but [30].

Answer: We thank the Reviewer for noticing this technical error, which has been corrected in the revised version.

2) It would be interested if authors can give more detailed description and commentary of the results of study of Seebeck coefficient of the sample.

Answer: Additional explanation has been inserted into the text.

“Changes in the value of the Seebeck coefficient with temperature for the oxide with y=0.7 measured in our work are in between the results reported earlier [30, 43] for the unsubstituted cobaltite and ferrite and are in good agreement with the proposed conductivity model. At T< 600 the conductivity increases mainly due to a raise in the mobility of charge carriers while their concentration remains unchanged. Thus, according to the Heikes formula [45] the value of the Seebeck coefficient should decrease or remain unchanged. At T> 600 K, the concentration of charge carriers decreases with temperature (eq. (4)) the Heikes formula [45] predicts an increase in the Seebeck coefficient. It is worth noting that the charge disproportionation process will also affect the concentration of 3d metals in various oxidation states, so a more detailed discussion of this issue is not possible.”

Reviewer 3 Report

The authors investigate the effect of element concentration on semiconductor structural parameters. The Seebeck coefficient indicates the predominant p-type conductivity of the formed semiconductors, and the oxygen content is determined by thermogravimetric analysis. The influence of the degree of Fe-substitution on the homogeneity range, crystal structure, oxygen nonstoichiometry, and electrical properties of Sr-enriched Gd1-xSrxCo1-yFeyO3-. The manuscript requires only minor revisions. My thoughts are as follows:

·        What are the values of δ on page 1 line 7?

·        Why is the Seebeck coefficient showing two trends? (Figure 9b)

·        Other structural parameters such as XRD EDX mapping, XPS, and so on could be used to confirm the oxygen ratio in the composites.

·        It is unclear why the Fe atoms have such an impact on the oxygen content.

Author Response

The authors would like to thank the Reviewers for their valuable comments that aimed to improve the manuscript at the peer-review stage. All Reviewers’ comments have been considered and required corrections were made. All corrections in the revised manuscript are marked with yellow highlighting.

1) What are the values of δ on page 1 line 7?

Answer: There symbol δ illustrates the value of oxygen nonstoichiometry that is changing within the wide range with temperature depending on a particular composition (x, y). The particular value of δ for each sample can be found in the Fig. 7.

2) Why is the Seebeck coefficient showing two trends? (Figure 9b)

Answer: Additional explanation has been inserted into the text.

“Changes in the value of the Seebeck coefficient with temperature for the oxide with y=0.7 measured in our work are in between the results reported earlier [30, 43] for the unsubstituted cobaltite and ferrite and are in good agreement with the proposed conductivity model. At T< 600 the conductivity increases mainly due to a raise in the mobility of charge carriers while their concentration remains unchanged. Thus, according to the Heikes formula [45] the value of the Seebeck coefficient should decrease or remain unchanged. At T> 600 K, the concentration of charge carriers decreases with temperature (eq. (4)) the Heikes formula [45] predicts an increase in the Seebeck coefficient. It is worth noting that the charge disproportionation process will also affect the concentration of 3d metals in various oxidation states, so a more detailed discussion of this issue is not possible.”

3) Other structural parameters such as XRD EDX mapping, XPS, and so on could be used to confirm the oxygen ratio in the composites.

Answer: We agree that many methods can be used for the determination of oxygen content in oxide materials. The advantage of TGA is that it allows to determine the oxygen content in bulk materials. The methods that were mentioned by the Reviewer are more efficient for examination of surface layers, which sometimes differ from the bulk. Therefore, we believe that TGA is one of the precise methods for such oxides.

4) It is unclear why the Fe atoms have such an impact on the oxygen content.

Answer: The explanation for such an impact on the oxygen content is given on page 7 :” Since iron is more electropositive element compared to cobalt (XFe=1.64 and XCo=1.7 [42]) it acts as an electron donor, preventing formation of oxygen vacancies.”